# Genotypic Profile and Clinical Characteristics of CRX-Associated Retinopathy in Koreans

**DOI:** 10.3390/genes14051057

**Published:** 2023-05-08

**Authors:** Dong Geun Kim, Kwangsic Joo, Jinu Han, Mihyun Choi, Seong-Woo Kim, Kyu Hyung Park, Sang Jun Park, Christopher Seungkyu Lee, Suk Ho Byeon, Se Joon Woo

**Affiliations:** 1Department of Ophthalmology, Seoul National University College of Medicine, Seoul National University Bundang Hospital, Seongnam 13620, Republic of Korea; kidogu@naver.com (D.G.K.); joo_man@hanmail.net (K.J.); jiani4@snu.ac.kr (K.H.P.); sangjunpark@snu.ac.kr (S.J.P.); 2Department of Ophthalmology, Inje University College of Medicine, Busan Paik Hospital, Busan 47392, Republic of Korea; 3Institute of Vision Research, Department of Ophthalmology, Yonsei University College of Medicine, Severance Hospital, Seoul 06273, Republic of Korea; jinuhan@yuhs.ac (J.H.); sklee219@yuhs.ac (C.S.L.); 4Department of Ophthalmology, Guro Hospital, Korea University College of Medicine, Seoul 08308, Republic of Korea; mnyoung23@gmail.com (M.C.); ksw64723@korea.ac.kr (S.-W.K.)

**Keywords:** CRX, cone-rod dystrophy, macular dystrophy, retinitis pigmentosa, Leber congenital amaurosis, Korean population

## Abstract

This study aimed to investigate the clinical characteristics of Korean patients with retinal dystrophy associated with pathogenic variants of cone rod homeobox-containing gene (CRX). We retrospectively enrolled Korean patients with CRX-associated retinal dystrophy (CRX-RD) who visited two tertiary referral hospitals. Pathogenic variants were identified using targeted panel sequencing or whole-exome sequencing. We analyzed clinical features and phenotypic spectra according to genotype. Eleven patients with CRX-RD were included in this study. Six patients with cone-rod dystrophy (CORD), two with macular dystrophy (MD), two with Leber congenital amaurosis (LCA), and one with retinitis pigmentosa (RP) were included. One patient (9.1%) had autosomal recessive inheritance, and the other ten patients (90.9%) had autosomal dominant inheritance. Six patients (54.5%) were male, and the mean age of symptom onset was 27.0 ± 17.9 years. At the first presentation, the mean age was 39.4 ± 20.6 years, and best-corrected visual acuity (BCVA) (logMAR) was 0.76 ± 0.90 in the better eye. Negative electroretinography (ERG) was observed in seven (63.6%) patients. Nine pathogenic variants were identified, including two novel variants, c.101-1G>A and c.898T>C:p.(*300Glnext*118). Taken together with the variants reported in prior studies, all variants within the homeodomain are missense variants, whereas most variants downstream of the homeodomain are truncating variants (88%). The clinical features of pathogenic variants within the homeodomain are either CORD or MD with bull’s eye maculopathy, whereas variants downstream of the homeodomain cause more diverse phenotypes, with CORD and MD in 36%, LCA in 40%, and RP in 24%. This is the first case series in Korea to investigate the CRX-RD genotype–phenotype correlation. Pathogenic variants downstream of the homeodomain of the CRX gene are present as RP, LCA, and CORD, whereas pathogenic variants within the homeodomain are mainly present as CORD or MD with bull’s eye maculopathy. This trend was similar to previous genotype–phenotype analyses of CRX-RD. Further molecular biologic research on this correlation is required.

## 1. Introduction

The cone rod homeobox-containing gene (CRX; OMIM *602225), located on chromosome 19q13.33, encodes a transcription factor crucial for the development and survival of photoreceptors [1,2,3]. It was first discovered in 1994 to be associated with autosomal dominant cone-rod dystrophy (CORD2; OMIM #120970) [4]. It contains four exons encoding a 299-amino-acid protein with a homeodomain near its N-terminus [3]. Mutations in the CRX gene can lead to diverse retinal phenotypes. To date, 139 pathogenic CRX variants have been associated with a wide range of inherited retinal dystrophy (IRD) phenotypes, including cone-rod dystrophy (CORD), Leber congenital amaurosis (LCA), retinitis pigmentosa (RP), and macular dystrophy (MD). In addition, most reports have demonstrated that this disorder is inherited as an autosomal dominant pattern (The Human Gene Mutation Database; http://www.hgmd.cf.ac.uk, accessed on 10 March 2022).

CRX-associated retinal dystrophy (CRX-RD) is relatively rare, accounting for 1.7% of LCA [5], 1.8% of CORD [6], and 1.7% of non-syndromic RP cases [7]. Since the report by Hull et al. [8], several genotype-oriented analyses of CRX-RD have been reported [9,10,11]. Because of its rarity, further genotype–phenotype research is still needed to understand the natural history and pathogenetic mechanism related to its diverse phenotypes. Although genetic defects in several genes such as ABCA4, RP1 and PROM1 cause diverse retinal phenotype, CRX is also one of those genes that can be expressed in the above four major phenotypes [12]. Therefore, genotype–phenotype correlation analyses can help to expand our knowledge to understand the mechanisms of various retinal phenotypes of IRDs. However, the evidence of genotype–phenotype correlation is still insufficient due to a small number of studies, and there is no report of CRX-RD among Korean patients. Therefore, we conducted a genotype–phenotype analysis of CRX-RD in Korean patients.

## 2. Materials and Methods

### 2.1. Patients and Clinical Assessment

We included all Korean patients with CRX-RD who visited two tertiary hospitals, Seoul National University Bundang Hospital and Severance Hospital, between September 2013 and October 2022. This study was approved by the Institutional Review Board (IRB) of Seoul National University Bundang Hospital (IRB no. B-1105-127-014 and no. B-2107-695-101) and adhered to the tenets of the Declaration of Helsinki. Informed consent was obtained from all patients before genetic analyses.

Comprehensive patient history was recorded, including the types and onset of symptoms as well as family history. Age of symptom onset was calculated based on the onset time of the first symptoms. Additionally, patients underwent standard ophthalmic examinations, including measurements of best-corrected visual acuity (BCVA), refractive errors, fundus photography, spectral-domain optical coherence tomography (SD-OCT; Spectralis OCT; Heidelberg Engineering, Heidelberg, Germany), and full-field standard electroretinogram (ERG). Static visual field examination, fundus autofluorescence (FAF), and multifocal ERG were performed in some patients. Full-field ERG was performed using procedures based on the International Society for Clinical Electrophysiology of Vision (ISCEV) [13,14,15,16].

### 2.2. Genetic Analyses

A customized target enrichment kit (Celemics, Seoul, Republic of Korea) was designed to cover the exon and splicing regions of 254 IRD-associated genes [17,18]. The captured library was sequenced using an Illumina NextSeq550 instrument (Illumina, San Diego, CA, USA) to generate 2 × 150 bp reads. Alignment to the hg19 human genome (BWA-MEM), post-alignment, and recalibration (‘Picard’ ver1.115 and ‘GATK’ ver4.0.4.0.), variant calling (GATK HaplotypeCaller), and annotation (ANNOVAR 2019Oct24). 

The clinical significance of each variant was categorized according to the latest recommendations of the American College of Medical Genetics and Genomics standards [19,20]. The Genome Aggregation Database (gnomAD), CADD, PolyPhen-2, and SIFT were used to exclude common variants and to identify disease-causing variants.

### 2.3. Clinical Subgroups

For this study, clinical subgroups were defined based on clinical features and electrophysiological examinations, and definitions from previous studies of CRX-RD were used [8]. LCA was defined as clinical findings within six months of age, accompanied by nondetectable ERG, and CORD was defined as progressive retinal dystrophy and greater cone dysfunction than rods on ERG, retinitis pigmentosa (RP) or rod-cone dystrophy (RCD) with progressive retinal dystrophy and greater rod dysfunction than that caused by ERG. Cone dystrophy (COD) was defined as progressive retinal dystrophy with only cone dysfunction on ERG, and MD was defined as normal full-field ERG with macular dysfunction.

## 3. Results

Eleven patients with nine variants of the CRX gene were included in this study. Of these, two patients (patients 2 and 8) had been previously reported [21,22]. One patient (9.1%) had autosomal recessive inheritance with compound heterozygosity (patient 2, c.101-1G>A/c.122G>A), four patients (36.4%) had autosomal dominant inheritance, and the other six patients (54.5%) were sporadic cases (molecularly autosomal dominant inheritance). Four benign or likely benign variants were excluded from the analysis.

### 3.1. Phenotypic Features of Patients

The clinical information and laboratory findings of the included patients are summarized in Table 1. Six patients (54.5%) were male, and the mean age of symptom onset was 27.0 ± 17.9 years (range, ~50 years). In three patients (27.3%), symptoms started before the age of four years; two patients (66.7%) were diagnosed with LCA and one patient (33.3%) with CORD. The remaining eight patients (72.7%) had symptoms after 25 years of age. At the first presentation, the mean age was 39.4 ± 20.6 years, and BCVA (logMAR) was 0.76 ± 0.90 for the better eye. Three patients had hyperopia, and five of the eight myopic patients had high myopia (more than −6.0D).

Six patients were diagnosed with CORD (54.5%), two with MD (18.2%), one with RP (9.1%), and two with LCA (18.2%). In addition, electronegative ERG findings were observed in seven patients (63.6%). The demographics and clinical features of all patients are summarized in Table 1. The representative cases of each phenotype are presented in Figure 1, and multimodal images of all included patients are presented in the Appendix A. Eight patients diagnosed with CORD or MD showed varying degrees of macular degeneration and visual acuity, but characteristic bull’s eye maculopathy was clearly observed in all six patients (100%) who underwent FAF. In the correlation analysis, age at symptom onset and BCVA (logMAR) were negatively correlated (*p* < 0.01). In addition, the refraction error and BCVA (logMAR) showed a positive correlation (the more myopic, the better visual acuity) (*p* = 0.01), but this result includes the extreme values of two LCA patients with hyperopia. Therefore, there is a limitation in interpreting it as a linear correlation. The distribution and correlation of symptom onset, BCVA, and refractive error are analyzed in Figure 2. 

### 3.2. Genotypes of Patients

Nine CRX variants were identified (c.101-1G>A, c.122G>A, c.118C>T, c.121C>T, c.128G>A, c.193G>C, c.193G>C, c.442delG, c.684G>C, c.898T>C (NM_000554.5)). Genetic profiles and in silico molecular analysis are summarized in Table 2. All patients had heterozygous mutations and one had compound heterozygosity (Patient 2, c.101-1G>A, c.122G>A). Seven variants have been previously reported [9,22,23,24,25,26]. All five mutations within the homeodomain were missense mutations. The three mutations in the homeodomain were truncating, missense, and stoploss mutations.

Two novel variants were identified in patient 1 (c.101-1G>A) and patients 10 and 11 (c.898T>C), and they were found to be sporadic. The c.101-1G>A mutation is a splicing variant immediately before exon 3, where the homeodomain starts (Figure 3). It has been reported as one of two compound heterozygous variants [21]. Previously, a heterozygous variant of c.100+2T>C (splicing donor), which may cause a molecularly similar effect to the c.101-1G>A mutation (splicing acceptor), has been reported to causes bull’s eye maculopathy [27]. The c.898T>C induces stoploss variant, resulting in amino acid changes in p.(*300Glnext*118). In a previous report, c.899A>G:p.(*300Trpext*118), which induces a similar type of stoploss variant, was reported as a pathogenic variant [28]. Taken together with the phenotypic variability of CRX-RD, the c.101-1G>A and c.898T>C variants can be sufficiently causative of retinopathy.

### 3.3. Genotype–Phenotype Correlation

Because of the limited number of patients, genotype–phenotype correlations were analyzed using pooled data from our study and four prior studies on CRX-RD. The genotype–phenotype analyses of the pooled data are summarized in Table 3, and a total of 34 reported pathogenic variants (83 affected patients) from 5 studies are summarized in Table 4. Of the 34 variants, 8 were variants within the homeodomain and 25 were variants after the homeodomain; the other variant was a splicing variant before the homeodomain (c.101-1G>A). All eight variants in the homeodomain were missense mutations (100%). In contrast, among the 25 variant mutations after the homeodomain, 22 were truncating mutations (88%), two were missense mutations (8%), and the other was a stoploss mutation (4%). Five hotspots were reported in two or more ethnic groups, all of which were within the homeodomain. Clinical features by phenotype in the collected data of 83 patients are shown in Table 5. The onset of symptoms was significantly faster in the order of LCA, RCD, CORD and MD (*p* < 0.01), and there was no significant difference between CORD and MD (*p* = 0.60). BCVA was significantly worse in LCA (*p* < 0.01), and there was no significant difference among the other three groups.

Among the 83 patients, only three patients showed an autosomal recessive pattern (one compound heterozygous patient (c.101-1G>A/c.122G>A) and two homozygous patients (c.193G>C)). These three patients showed more severe phenotypes (LCA and RP) than those with the same variants in an autosomal dominant pattern. In addition, six of the eight mutations within the homeodomain (75.0%) were diagnosed as CORD or MD dominantly. All patients with pathogenic variants within the homeodomain showed CORD or MD phenotypes, except the aforementioned three patients with an autosomal recessive pattern. In contrast, 25 mutations downstream of the homeodomain showed various phenotypes. Nine variants (36%) were diagnosed as CORD or MD, ten variants (40%) were diagnosed as LCA, and six variants (24%) were diagnosed as RP (Table 4 and Figure 3).

Ethnic differences can be found in Table 6. However, as the number of patients and studies is small, a careful interpretation is required. For example, in the Chinese study, the proportion of LCA was very high. We suspect that this might be due to the subspecialty of the investigators or the characteristics of the general patients that visited the research institution. Another possibility is that some variants might have been over-estimated due to the inclusion of related patients in some studies. Therefore, further research using larger number of patients is needed to reveal the ethnic differences.

## 4. Discussion

In this study, we identified nine pathogenic variants among eleven Korean patients with *CRX*-RD. Two novel variants (c.101-1G>A and c.898T>C) were found in two unrelated patients. Variants within the homeodomain were homogeneously diagnosed as CORD or MD with bull’s eye macular degeneration, except in one patient with compound heterozygosity. In contrast, variants downstream of the homeodomain can present with variable retinal dystrophy including CORD, LCA, and RP.

Recently, Nishiguchi et al. analyzed the CRX-RD of 10 Japanese families and concluded that variants within or downstream of the homeodomain are associated with different phenotypes [11]. Although they used a different phenotypic classification, they reported a more dominant phenotypical distribution of “retinal degeneration with macular degeneration” in variants within the homeodomain. The genotype–phenotype correlation in our Korean patients was similar to this result. Thus far, a genotype-oriented analysis of CRX-RD has been performed in four studies [8,9,10,11]. When analyzing 34 mutations in 83 patients reported in these studies, including our study, there seemed to be clear genotypic and phenotypic differences according to the location of the mutation. Most patients with simple heterozygous variants in the homeodomain are predominantly diagnosed with CORD or MD. However, among the 25 mutations located downstream of the homeodomain, CORD or MD was observed in only 36%, LCA in 40%, and RP in 24%. Previously, such a genotype–phenotype association was not observed in a comprehensive review by Hull or Rivolta [8,31]. This might be an ethnic difference, but at that time, there were significantly fewer reports of CRX-RD than now. Another possibility is that consistent phenotypic classification was not applied because most previous reports on CRX variants have focused on causative genes in specific phenotypes.

One of the known characteristics of pathogenic CRX variants is their peculiar distribution of the mutation type. Most of missense variants are located within the homeodomain, and most truncating variants are located downstream of the homeodomain, which was first recognized by Rivolta et al. [31]. This phenomenon is thought to be because the truncating mutation located in the last exon after the homeodomain may avoid nonsense-mediated decay (NMD) [36]. Yi et al. also reported that truncating variants before the homeodomain are likely benign, and many missense variants outside the domains are likely benign [9]. In our study, all five mutations within the homeodomain were missense mutations, and most missense mutations (4/5, except c.684G>C) after homeodomains were determined to be benign or likely benign variants. This is consistent with previous results. 

CRX is a paired-homeodomain transcription factor, essential for photoreceptor and bipolar cell transcriptional networks during differentiation and cell fate determination in the vertebrate retina [3,37]. Mammalian CRX protein is 299 amino acids in length. Its functional domain, except for the homeodomain, has been described differently in various papers, but most recently, it has been described as the homeodomain and the transcription factor Otx (164-249aa) (Ensembl; https://www.ensembl.org/, accessed on 22 March 2023). The homeodomain is responsible for DNA binding function in paired form [38,39,40], and the C-terminal portion is known to be involved in transcriptional regulation [41,42]. Considering that carriers of heterozygous whole CRX deletion showed normal retinal findings in previous reports [43,44] we assumed that truncating mutations within or before the homeodomain do not show pathological changes because of NMD, while missense mutations within the homeodomain induce pathological changes by interacting (interfering) with wildtype CRX in the DNA binding process. However, a recent study reported that single CRX whole-gene deletion also causes dominant late-onset macular disease [45]; therefore, further research on this molecular mechanism is required. Mutations within the transactivation domain have no restrictions on the type of mutation due to NMD escape, but mutations that cause large changes, such as truncating mutations, appear to be necessary for most pathological changes [46]. These mutations can appear much more diverse than single-amino-acid substitutions, which may be the cause of more diverse clinical features in the mutations of the transactivation domain than in mutations of the homeodomain in our genotype–phenotype analysis.

Electronegative ERG was present in most of the patients in our study. This result is consistent with that reported by Nishiguchi et al. [11]. In their study, electronegative ERG was present in all detectable ERG and was the only ophthalmic abnormality in asymptomatic family members. However, in our patients, only one patient with macular dystrophy did not prove this finding, so there was a phenotypic difference. Nevertheless, it is clear that electronegative ERG frequently accompanies CRX-RD, suggesting the involvement of not only photoreceptors but also bipolar cell transcriptional networks.

This study has several limitations. First, we included CRX-RD patients from two tertiary hospitals, but only a limited number of eleven patients with nine pathogenic variants were included in the study. These numbers are too small to represent the entire clinical features of CRX-RD in Korea, so we performed a genotype–phenotype correlation analysis together with previous reports from other countries. In addition, many patients had sporadic family histories, and even in several symptomatic family members, a comprehensive ophthalmic examination could not be performed due to their refusal. Therefore, it was difficult to observe intrafamilial variation in our study. However, three variants (c.101G>A, c.193G>C, and c.898T>C) were commonly found in unrelated patients. These patients show different clinical features indicating the phenotypic variability. Finally, data on long-term prognosis were insufficient. Nevertheless, our study has several strengths. To our knowledge, this is the first report of a genotype-oriented case series of CRX-RD with various phenotypes, and eleven patients with nine pathogenic variants form the largest cohort of CRX-RD reported in Korea. In addition, we identified two novel variants (c.101-1G>A and c.898T>C) found in two unrelated Korean patients.

## 5. Conclusions

This is the first Korean CRX-RD case series to investigate the genotype–phenotype correlation. Mutations downstream of the homeodomain of the CRX gene are present as RP, LCA, and CORD, whereas mutations within the homeodomain are mainly present as CORD or MD with bull’s eye maculopathy. This trend was similar to previous genotype–phenotype analyses of CRX-RD. Further molecular biologic studies on this correlation are needed.

## Figures and Tables

**Figure 1 genes-14-01057-f001:**
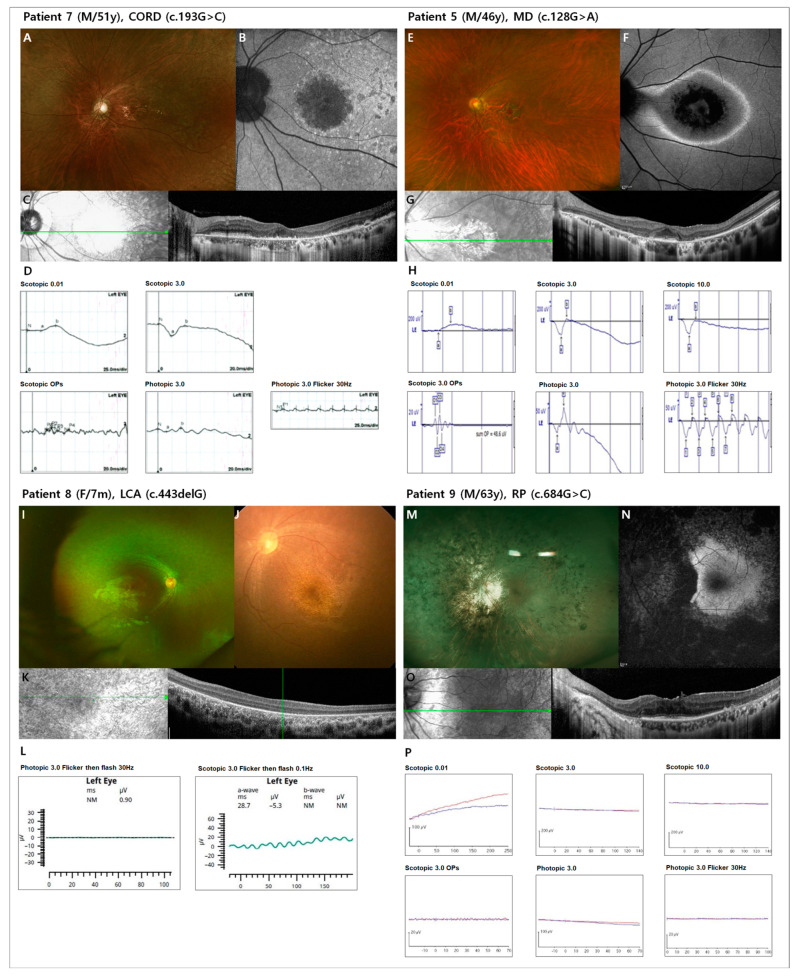
Color fundus photography, optical coherence tomography (OCT), fundus autofluorescence (FAF) and electroretinogram (ERG) of representative cases by each phenotype. In patient 7 (**A**–**D**, c.193G>C, cone-rod dystrophy (CORD)), round macular atrophy and peripheral granularity were observed in fundus photography and FAF (**A**,**B**); subfoveal atrophy and parafoveal blurring of photoreceptor layers were observed in OCT (**B**); and cone-dominant reduction in response observed in ERG (**D**). In patient 5 (**E**–**H**, c.128G>A, macular dystrophy (MD)), Perifoveal ring-shaped retinal pigment epithelium atrophy was observed in fundus photography (**E**) and OCT (**G**). Outside of the atrophic lesions, ring-shaped hyperautofluorecence was observed in FAF (**F**). Photoreceptor response in ERG was normal (**H**). In patient 8 (**I**–**L**, c.443del, Leber congenital amaurosis (LCA)) and FAF were not performed, and only color fundus photographs with good quality were attached. Blond fundus and white punctate lesions in peripheral retina (**I**,**J**) and diffuse blurring of photoreceptor layers (**K**) were observed. In patient 9 (**M**–**P**, c.684G>C, retinitis pigmentosa (RP)), peripheral retinal degeneration with bone spicule (**M**,**N**) and parafoveal sparing of retinal pigment epithelium atrophy (**O**) were observed. ERG was extinguished in patients 8 and 9 (**L**,**P**) and electronegative ERG was observed in patients 7 and 5 (**D**,**H**).

**Figure 2 genes-14-01057-f002:**
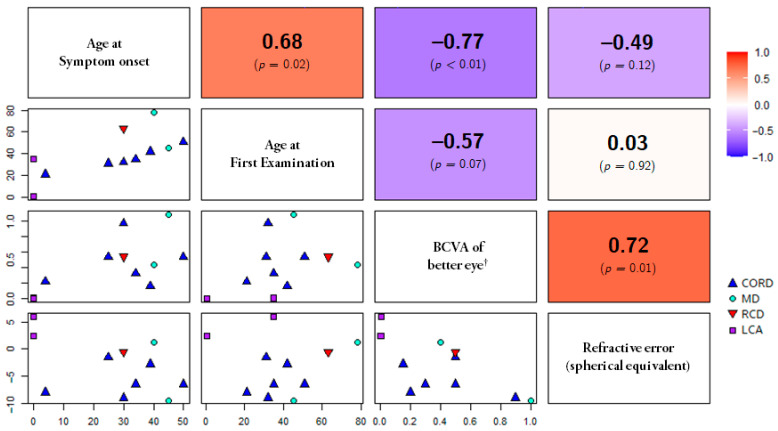
Distribution and correlation analysis between age, best-corrected visual acuity, and refractive error of patients. The six plots on the lower left are the distribution of each patient (Blue triangle is cone-rod dystrophy, light blue circle is macular dystrophy, red inverted triangle is rod-cone dystrophy, and purple square is Leber congenital amaurosis). The six plots on the upper right showed the degree of correlation between each variable (The closer to red, the stronger the positive correlation; the closer to purple, the stronger the negative correlation; and bold letters are the correlation coefficients). Age at symptom onset and BCVA (logMAR) showed a negative correlation (*p* < 0.01). Age at symptom onset and age at first examination showed a positive correlation (*p* = 0.02). In addition, refraction error and BCVA (logMAR) showed a positive correlation (*p* = 0.01), but this result includes the extreme values of two LCA patients with hyperopia, so there is a limitation in interpreting it as a linear correlation. **^†^** Please note that decimal BCVA was used for distribution plots for the convenience of the reader while logMAR visual acuity was used for correlation analysis. BCVA, best-corrected visual acuity; CORD, cone-rod dystrophy; LCA, Leber congenital amaurosis; MD, macular dystrophy; RP, retinitis pigmentosa.

**Figure 3 genes-14-01057-f003:**
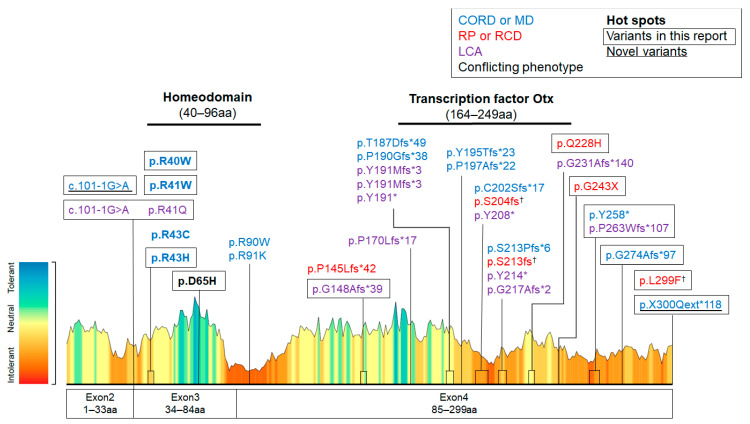
Schematic diagram of *CRX* structure and phenotype distribution of reported variants from this paper and four previous studies on *CRX*-associated retinopathy. A total of 35 variants from 83 affected patients were included (cone-rod dystrophy (CORD) or macular dystrophy (MD) in blue letters, retinitis pigmentosa (RP) or rod-cone dystrophy (RCD) in red letters, Leber congenital amaurosis (LCA) in purple letters, and conflicting phenotype in black letters). Hot spots commonly reported in two or more studies are indicated in bold. All mutations within the homeodomain were missense mutations, and most showed phenotypes of CORD or MD. Notably, few patients with other phenotypes were either compound heterozygosity (c.101-1G>A/c.122G>A(p.R41Q), LCA) or homozygosity (p.D65H, RP), unlike all the other patients were simple heterozygosity. On the other hand, the majority of the mutations after the homeodomain were truncating mutations (88%), and various phenotypes appeared. (CORD and MD in 36%, LCA in 40%, and RP in 24%). † This study used a different phenotypic classification system. We performed phenotypic classification based on the test results provided.

**Table 1 genes-14-01057-t001:** Demographics and clinical features of patients.

Patient No.	CRX Variants	Molecularly Raised Inheritance	Inheritance Based on Family History	Sex	Age at Onset (years)	Age at Examination (years)	Symptom	Initial Decimal BCVA (logMAR Unit)		Refractive Errors		Phenotype Subgroup	Fundus Appearance	ERG	VF
								Right	Left	Right	Left				
1	c.101-1G>A	AD	Sporadic	F	34	35	Decreased vision	0.2 (0.7)	0.3 (0.52)	−6.00	−7.00	CORD	Mild RPE irregularity in macula	More prominent reduction in cone response, negative ERG	Paracentral ring scotoma
2	c.101-1G>A c.122G>A	AR	AR	M	Birth	35	Decreased vision	HM	HM	5.50	6.50	LCA	Central macular atrophy, peripheral retinal atrophy, bone spicules	Undetectable	N/A
3	c.118C>T	AD	Sporadic	F	25	31	Decreased vision	0.5 (0.3)	0.4 (0.4)	−1.50	−1.50	CORD	Mild RPE irregularity in macula, annular depigmentation along the arcade	More prominent reduction in cone response, negative ERG	Paracentral ring scotoma
4	c.121C>T	AD	AD	F	4	21	Decreased vision	0.2 (0.7)	0.2 (0.7)	−8.50	−9.50	CORD	Mild RPE irregularity in macula	Severe reduction in cone response, negative ERG	Central scotoma
5	c.128G>A	AD	AD	M	45	46	Visual distortion	1.0 (0.0)	1.0 (0.0)	−9.5	−9.50	MD	Ring of RPE atrophy in macula	Negative ERG	Cecocentral scotoma
6	c.193G>C	AD	Sporadic	M	40	78	Decreased vision, photophobia	0.3 (0.52)	0.4 (0.4)	1.25	1.25	MD	Ring of RPE atrophy in macula, mild granularity in peripheral retinae	Normal ERG	N/A
7	c.193G>C	AD	Sporadic	M	50	51	Decreased vision	HM	0.5 (0.3)	−6.00	−7.00	CORD	Round RPE atrophy in macula, moderate granularity in peripheral retinae	Severe reduction in cone response, negative ERG	Central scotoma
8	c.443del	AD	AD	F	Birth	7 mo	Poor eye contact	USCM	USCM	2.50	2.50	LCA	Blond fundus, peripheral white punctate lesions	Undetectable	N/A
9	c.684G>C	AD	AD	M	30	63	Night blindness	0.5 (0.3)	0.4 (0.4)	−0.50	−0.50	RP	Peripheral retinal atrophy, bone spicule, central atrophy of right macula, attenuated arterioles	Undetectable	Central tunnel
10	c.898T>C	AD	Sporadic	M	39	42	Decreased vision	0.15 (0.82)	0.15 (0.82)	−2.50	−3.00	CORD	RPE irregularity in macula	More prominent reduction in cone response, negative ERG	Central scotoma
11	c.898T>C	AD	Sporadic	M	30	32	Visual disturbance	0.9 (0.05)	0.8 (0.10)	−9.00	−9.00	CORD	RPE irregularity in macula	More prominent reduction in cone response, negative ERG	N/A

AD, autosomal dominant; AR, autosomal recessive; BCVA, best-corrected visual acuity; CORD, cone-rod dystrophy; ERG, electroretinogram; HM, hand movements; LCA, Leber congenital amaurosis; MD, macular dystrophy; N/A, not applicable; RP, retinitis pigmentosa; RPE, retinal pigment epithelium; USCM, no constant, no steady, no maintained fixation; VF, visual field examination.

**Table 2 genes-14-01057-t002:** Profiles and in silico molecular genetic analysis of nine variants in included patients.

Pt	Nucleotide	Amino Acid	CADD	Polyphen-2	SIFT	gnomAD	ClinVar	ACMG	Novel Variant
1	c.101-1G>A	-	33	NA	NA	NF	No interpretation for the single variant	P	Surl [21] However, novel as solitary
2	c.101-1G>A c.122G>A	- p.(Arg41Gln)	33 28.5	NA 0.998	NA 0	NF 1/31396	P P	P LP	Surl [21] Swain [23]
3	c.118C>T	p.(Arg40Trp)	28.9	0.996	0	1/250660	P	P	Arai [24]
4	c.121C>T	p.(Arg41Trp)	25.6	0.996	0	2/251204	P	P	Swain [23]
5	c.128G>A	p.(Arg43His)	28.4	0.995	0	1/251318	P	P	Yi [9]
6	c.193G>C	p.(Asp65His)	26.6	0.998	0	NF	P	LP	Jin [25]
7									
8	c.443del	p.(Gly148Alafs*39)	28.3	NA	NA	NF	P	P	Han [22]
9	c.684G>C	p.(Gln228His)	16.75	0.917	0.04	2/151064	LP	US	Jespersgaard [26]
10	c.898T>C	p.(*300Glnext*118)	16.07	NA	NA	NF	LP	LP	Novel
11									

Pt = patient, NA = Not applicable, CADD (https://cadd.gs.washington.edu/snv; accessed date: 23 November 2021), Polyphen-2 (http://genetics.bwh.harvard.edu/pph2; accessed date: 23 November 2021), SIFT (https://sift.bii.a-star.edu.sg/www/Extended_SIFT_chr_coords_submit.html; accessed date: 23 November 2021), ACMG = American College of Medical Genetics and Genomics. NM_000554.5 transcript was used.

**Table 3 genes-14-01057-t003:** Summary of previous studies and ours on the genotype–phenotype correlation of *CRX*-associated retinopathy.

Author	Publication Year	Number of Variants and Patients	Nationality	Genotype–Phenotype Correlation
Hull et al. [8]	2014	10 variants of 18 patients	UK	No evident association between age of onset and position or type of *CRX* mutation.
Yi et al. [9]	2019	12 pathogenic variants of 18 affected patients (total 24 variants including benign variants)	China	Approximately half of heterozygous missense variants are likely benign, heterozygous truncating variants affecting the homeodomain are likely benign. Truncating mutations after the homeodomain are likely associated with a more severe phenotype.
Fujinami et al. [10]	2020	8 variants of 18 patients	Japan	There seems to be a trend between phenotype and genotype (subgroups considering mutation type and zygosity).
Nishiguchi et al. [11]	2020	6 variants of 21 patients	Japan	Heterozygous mutations within or downstream of the homeobox domain in *CRX* relate to the different retinal phenotypes
Our study	2022	9 variants of 11 patients (total 13 variants of 15 patients including benign variants)	Korea	All mutations within the homeodomain are missense mutations, and most are expressed as cone-rod dystrophy or macular dystrophy. Most variants after the homeodomain are truncating mutations, and the phenotypes are diverse

**Table 4 genes-14-01057-t004:** Pathogenic variants and phenotypes in this study and previous studies on *CRX*-associated retinopathy.

Nucleotide Change	Amino Acid Change	Reference	Nationality	Zygosity	Phenotype	N of Affected Cases	Age of Symptom Onset	First Report
c.101-1G>A	N/A	This study	Korea	Heterozygous	CORD	1	34	This study
c.101-1G>A c.122G>A	N/A p.Arg41Gln	This study	Korea	Compound heterozygous	LCA	1	Birth	[21] [23]
c.118C>T	p.Arg40Trp	[10]	Japan	Heterozygous	CORD	5	30, 35, 56, NA, NA	[24]
		[9]	China	Heterozygous	RP	1	NA	
		This study	Korea	Heterozygous	CORD	1	25	
c.121C>T	p.Arg41Trp	[10]	Japan	Heterozygous	CORD	3	60, NA, NA	[23]
		[11] ^†^	Japan	Heterozygous	CORD ^†^	11	29, 30, 34, 39, 40, 45, 47, 53, 54, 58, 71	
		[9]	China	Heterozygous	CORD	1	NA	
		This study	Korea	Heterozygous	CORD	1	4	
		[8]	UK	Heterozygous	RP	1	3.5	
				Heterozygous	MD	1	53	
c.127C>T	p.Arg43Cys	[10]	Japan	Heterozygous	CORD	2	75, 77	[6]
		[11] ^†^	Japan	Heterozygous	CORD ^†^	1	16	
		[9]	China	Heterozygous	CORD	1	36	
c.128G>A	p.Arg43His	[9]	China	Heterozygous	LCA	1	ECH	[9]
		[10]	Japan	Heterozygous	MD	2	31, 62	
		This study	Korea	Heterozygous	MD	1	45	
c.193G>C	p.Asp65His	[10]	Japan	Homozygous	RP	2	37, NA	[25]
		This study	Korea	Heterozygous	MD	1	40	
					CORD	1	50	
c.268C>T	p.Arg90Trp	[10]	Japan	Heterozygous	CORD	1	45	[29]
c.272G>A	p.Arg91Lys	[8]	UK	Heterozygous	MD	1	35	[8]
c.434del	p.Pro145Leufs*42	[10]	Japan	Heterozygous	RP	1	15	[10]
c.443del	p.Gly148Alafs*39	This study	Korea	Heterozygous	LCA	1	Birth	[22]
c.509del	p.Pro170Leufs*17	[9]	China	Heterozygous	LCA	2	ECH, ECH	[30]
c.557dup	p.Thr187Aspfs*49	[9]	China	Heterozygous	CORD	2	NA, NA	[9]
c.568_590del	p.Pro190Glyfs*38	[8]	UK	Heterozygous	CORD	3	12, 12, 14	[8]
c.570del	p.Tyr191Metfs*3	[8]	UK	Heterozygous	LCA	2	Birth	[8]
c.571del	p.Tyr191Metfs*3	[8]	UK	Heterozygous	LCA	1	3mo	[31]
c.573T>G	p.Tyr191*	[9]	China	Heterozygous	LCA	1	0.3	[9]
c.582del	p.Tyr195Thrfs*24	[8]	UK	Heterozygous	MD	1	42	[8]
c.590del	p.Pro197Alafs*22	[10]	Japan	Heterozygous	CORD	2	30, 45	[10]
c.605del	p.Cys202Serfs*17	[8]	UK	Heterozygous	MD	1	50	[8]
					COD	1	45	
c.615del	p.Ser206Profs*13	[11] ^†^	Japan	Heterozygous	RP ^†^	1	40	[32]
c.624T>G	p.Tyr208*	[8]	UK	Heterozygous	LCA	1	Birth	[5]
				Heterozygous	RP	1	6	
c.636del	p.Ser213Profs*6	[9]	China	Heterozygous	CORD	1	12	[33]
c.639del	p.Tyr214Ilefs*5	[11] ^†^	Japan	Heterozygous	RP ^†^	1	6	[11]
c.642T>G	p.Tyr214*	[9]	China	Heterozygous	LCA	2	0.4, ECH	[9]
c.650del	p.Gly217Alafs*2	[9]	China	Heterozygous	LCA	2	ECH, ECH	[34]
c.684G>C	p.Gln228His	This study	Korea	Heterozygous	RP	1	30	[26]
c.692del	p.Gly231Alafs*140	[9]	China	Heterozygous	LCA	2	ECH, ECH	[9]
c.727G>T	p.Gly243*	[11] ^†^	Japan	Heterozygous	CORD ^†^	2	68, 51	[11]
					RP ^†^	1	45	
c.774T>A	p.Tyr258*	[8]	UK	Heterozygous	MD	2	49, 50	[8]
					CORD	1	32	
c.787_790del	p.Pro263Trpfs*107	[9]	China	Heterozygous	LCA	2	0.4, NA	[9]
c.821del	p.Gly274Alafs*97	[8]	UK	Heterozygous	CORD	1	11	[8]
c.897G>C	p.Leu299Phe	[11] ^†^	Japan	Heterozygous	RP ^†^	1	31	[35]
c.898T>C	p.*300Glnext*118	This study	Korea	Heterozygous	CORD	2	30,39	This study

COD, cone dystrophy; CORD, cone-rod dystrophy; ECH, early childhood; LCA, Leber congenital amaurosis; MD, macular dystrophy; NA, not available; RCD, rod-cone dystrophy; RP, retinitis pigmentosa. ^†^ This study used a different phenotypic classification system. We performed phenotypic classification based on the test results provided.

**Table 5 genes-14-01057-t005:** Clinical features according to phenotype from data pooled in this study and previous studies on *CRX*-associated retinopathy.

	CORD (*N* = 44)	MD (*N* = 10)	RCD (*N* = 11)	LCA * (*N* = 18)	*p* Value
Age at symptom onset (years)	39.0 ± 18.8	45.7 ± 9.1	23.7 ± 16.2	0.1 ± 0.2	<0.01 ^†^
BCVA of better eye (logMAR unit)	0.6 ± 0.5	0.4 ± 0.4	0.8 ± 0.8	2.4 ± 0.6	<0.01 ^‡^

* Terms such as ‘early childhood’ or ‘birth’ used in previous studies all calculated the onset age of 0 years old. ^†^ There were significant differences in all groups except between CORD and MD. ^‡^ There were significant differences between LCA and the remaining three groups, respectively. CORD, cone-rod dystrophy; LCA, Leber congenital amaurosis; MD, macular dystrophy; RP, retinitis pigmentosa; RCD, rod-cone dystrophy.

**Table 6 genes-14-01057-t006:** Clinical features according to nationality from data pooled in this study and previous studies on *CRX*-associated retinopathy.

	China (*N* = 18)	Japan (*N* = 36)	Korea (*N* = 11)	UK (*N* = 18)	*p* Value
Age at symptom onset	3.8 ± 10.2	43.7 ± 17.3	29.7 ± 16.4	23.0 ± 20.8	<0.01 ^†^
BCVA of better eye (logMAR unit)	1.3 ± 1.0	0.5 ± 0.5	0.8 ± 0.9	1.3 ± 1.0	<0.01 ^‡^
Phenotype					N/A
COD or CORD	5 (27.8%)	27 (75.0%)	6 (54.5%)	6 (33.3%)	
MD	0	2 (5.6%)	2 (18.2%)	6 (33.3%)	
RP or RCD	1 (5.6%)	7 (19.4%)	1 (9.1%)	2 (11.1%)	
LCA	12 (66.7%)	0	2 (18.2%)	4 (22.2%)	

^†^ There was a significant difference between the China group and the other three groups, as well as the Japan and UK groups. ^‡^ There was a significant difference in the Japan–China and Japan–UK groups. COD, cone dystrophy; CORD, cone-rod dystrophy; LCA, Leber congenital amaurosis; MD, macular dystrophy; N/A, not applicable; RP, retinitis pigmentosa; RCD, rod-cone dystrophy.

## Data Availability

Data are contained within the article. The additional data presented in this study are available on request from the corresponding author.

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
