# Peer review of "Genotypic Profile and Clinical Characteristics of CRX-Associated Retinopathy in Koreans"

_genes, 2023, doi:10.3390/genes14051057_

Round 1

Reviewer 1 Report

As stated by yourself, CRX-retinopathy is quite rare, so clinical reports on this condition are very welcome and you did a great job comparing your patients with earlier reports (table 4). However, I have missed information on the clinical picture, which you could have easily provided, I assume. For instance,

1) you do not mention intrafamilial variation in your patients, which in my experience can be quite impressive and is of great clinical importance

2) you are very short on the presenting symptoms: decrease in visual acuity is mentioned most of the times, but no mention is made of color vision problems, photophobia or other more specific complaints (for instance in patient 11 with near normal vision)

3) patient 5 with normal visual acuity complained of visual distortion: was epiretinal fibrosis seen on OCT?

4) an electronegative ERG is a rather specific feature, seen in a.o. in juvenile retinoschisis, some forms of congenital night blindness and MAR (melanoma associated retinopathy). I was not aware of its occurence in CRX-retinopathy, but the authors do not mention this important clue at all in their discussion! Nishiguchi et al. (20200, referenced by the authors, enlightened me on this subject.

5) it is of no importance to average BCVA and age in 11 patients with 4 different conditions (CORD, RP, MD and LCA) as is done in the Introduction. The individual values are plotted in figure 2, which is very difficult to comprehend as a clinician/ophthalmologist. We are still using BCVA in decimals or Snellen in the clinic, so "translating" of the BCVA from logMAR in clinical designation is a lot to ask from the reader. So please, report BCVA also in decimals in such a small group of patients.

Reviewer 2 Report

The authors screened eleven patients for CRX gene mutations, identified 9 variants, and used pooled data to correlate genotype with phenotype in 80 patients.  The following major concerns are related to this study.

1.       CRX gene variants show variation in phenotype, such as the age of onset and other clinical phenotypes; it needs to be clarified how mean values were calculated for patients having diverse symptoms.

2.       The authors have described the data of only 11 patients in the abstract; what about the findings of data taken from the patients pool (80 patients.)

3.       The segregation of the variants is important, particularly for the novel variants.

4.       The patients from other studies belong to different ethnic groups, and the authors may discuss the other confounding factors influencing the phenotype.

5.       Authors may also add the Insilco functional  analysis of identified variants

Round 2

Reviewer 1 Report

Thank you for adding much more clinical data, which are of essential value for the clinical ophthalmologist!

Reviewer 2 Report

The manuscript is improved now, and suggestions are incorporated where possible.